# Rare Genetic Syndromes and Oral Anomalies: A Review of the Literature and Case Series with a New Classification Proposal

**DOI:** 10.3390/children9010012

**Published:** 2021-12-26

**Authors:** Claudia Salerno, Valeria D’Avola, Luca Oberti, Elena Almonte, Elena Maria Bazzini, Gianluca Martino Tartaglia, Maria Grazia Cagetti

**Affiliations:** 1Department of Biomedical, Surgical and Dental Science, University of Milan, Via Beldiletto 1, 20142 Milan, Italy; valeria.davola@unimi.it (V.D.); luca.oberti@unimi.it (L.O.); elena.almonte@unimi.it (E.A.); elenabazzini@fastwebnet.it (E.M.B.); gianluca.tartaglia@unimi.it (G.M.T.); maria.cagetti@unimi.it (M.G.C.); 2UOC Maxillo-Facial Surgery and Dentistry Fondazione IRCCS Cà Granda, Ospedale Maggiore Policlinico, University of Milan, 20100 Milan, Italy

**Keywords:** rare genetic syndromes, dento-oro-maxillofacial anomalies, oral abnormalities

## Abstract

Rare genetic syndromes, conditions with a global average prevalence of 40 cases/100,000 people, are associated with anatomical, physiological, and neurological anomalies that may affect different body districts, including the oral district. So far, no classification of oral abnormalities in rare genetic syndromes is present in the literature. The aim of this narrative review is to analyze literature on rare genetic syndromes affecting dento-oro-maxillofacial structures (teeth, maxillary bones, oral soft tissues, or mixed) and to propose a classification according to the detected oral abnormalities. In addition, five significant cases of rare genetic syndromes are presented. The Scale for the Assessment of Narrative Review Articles (SANRA) was followed for this review. From 674 papers obtained through PubMed search, 351 were selected. Sixty-two rare genetic syndromes involving oral manifestations were found and classified. The proposed classification aims to help the clinician to easily understand which dento-oro-maxillofacial findings might be expected in the presence of each rare genetic syndrome. This immediate framework may both help in the diagnosis of dento-oro-maxillofacial anomalies related to the underlying pathology as well as facilitate the drafting of treatment plans with the involvement of a multidisciplinary team.

## 1. Introduction

A genetic syndrome is a condition caused by any abnormality in one’s genome that may be inherited or de novo. There is no clear and unique definition of rare genetic syndrome, but the meaning most commonly attributed to it in literature is that of a pathology with an average prevalence threshold ranging between five and 76 cases/100,000 people, with a global average prevalence of 40 cases/100,000 people. The Orphan Drug Act defines a rare disease or condition as a disease (a) which affects less than 200,000 people in the United States or (b) for which there is no reasonable expectation that the cost of developing a drug and making it available in the US will be recovered from its sale [1]. The member states of the European Union have adopted as a definition of rare disease, “the life-threatening or chronically debilitating conditions that affect maximum 5 persons over 10,000” [2].

Genetic syndromes are associated with anatomical, physiological, and neurological differences that may affect different body districts, including the mouth and its associated structures. Chromosomal localization is known for more than 50 of genetic diseases and responsible mutated genes have been discovered in more than 60 of these diseases with craniofacial and oral phenotypes alterations [3]. According to the London Dysmorphology Database [4], in 2011, approximately 900 out of 5000 genetic syndromes include dental-oro-maxillofacial anomalies in their clinical pictures, mostly among those affecting the ectodermal derivatives. Understanding dental genetics and developmental biology will allow the identification of the processes involved in the genesis of specific abnormalities [5,6,7].

Warning oral signals are dental and oral anomalies which, together with extraoral signals, can facilitate the early diagnosis of rare genetic syndromes. Dental anomalies consist in alterations of the external appearance, internal structure, or topography of one or more primary or permanent teeth, resulting from a disorder that can be genetically determined, congenital, or acquired. Dental anomalies can include reduced or increased teeth number, as well as variations in shape or volume, location or position, development, and structure of teeth (Table 1).

Developmental dental defects can result from teratogenic actions during odontogenesis as well as genetic disorders, isolated or associated to other signs in a syndrome [8]. Dental malformations may be the most evident manifestations, and therefore first diagnosed, while other signs affecting different organs will only manifest themselves later. These anomalies have both functional and aesthetical implications [9].

Dentists should be aware of the existence of rare genetic syndromes and should know how to clinically recognize, manage, and treat the dento-oro-craniofacial anomalies they might include. Dentists, in fact, could be asked to cooperate with other specialists in the diagnosis of these diseases. Rare genetic syndromes, despite their generally chronic and progressive nature, have long-term complications that can be reduced or delayed if they are early diagnosed and managed [10]. An oral rehabilitation is frequently necessary to guarantee mastication, swallowing, phonetics, and mechanical ventilation. Moreover, good dental appearance can positively contribute to the integration of affected subjects in society as well in school [11].

The main sources of information for clinicians on management recommendations for patients with rare genetic syndromes are very limited and consist of: Internet sites such as Orpha.net [12], a portal for rare diseases and orphan drugs, which offers a list of the diseases; databases such as Phenodent, where it is possible to find help in the diagnosis and management of dental anomalies [13]; the Genetic and rare diseases information center (GARD) [14], a program of the National Center for Advancing Translational Sciences, where the rare diseases are classified and explained; and the Online Mendelian Inheritance in Man (OMIM) [15,16], that since 1966 has continuously provided an updated catalog of human genes and genetic disorders and traits, and it has been online since 1987. Orpha.net provides an exhaustive list of all rare diseases, updated every year.

Rare diseases present on Orpha.net are defined according to two criteria:Homogeneity: Each disease is defined by its clinical homogeneity, regardless of the etiology or the number of identified responsible genes.Rarity: Each disease is defined based on the European legislation, which establishes a prevalence threshold not exceeding 5 affected persons per 10,000 people.

Rare diseases need to be described in the international scientific literature (peer-reviewed articles) with at least two cases, confirming the non-random association of clinical signs [17].

Regardless of the oral malformation observed, therapeutic management is often long and complex. It starts at the time of diagnosis and continues during the child’s growth until adulthood, with the involvement of a multidisciplinary team.

The aim of this narrative review is to search through literature cases of rare genetic syndromes affecting the cranio-facial structures (teeth, maxillary bones, oral soft tissues or mixed) and to propose a classification of oral abnormalities in rare genetic syndromes. In addition, five significant cases of rare genetic syndromes are presented. The clinical cases presented have been referred to the Dental Clinic of San Paolo Hospital in Milan (Italy) since they have required a challenging multidisciplinary diagnostic and therapeutic approach.

## 2. Materials and Methods

A narrative review of the literature was conducted to identify relevant studies. This review article follows the Scale for the Assessment of Narrative Review Articles (SANRA) (Appendix A) criteria. Literature describing rare genetic syndromes and dento-oro-craniofacial anomalies was identified by searching through PubMed. The literature search was carried out including papers published from January 2000 to July 2021 by S.C and D.V. The following search string was used: Rare genetic syndromes AND ((oral OR dental OR maxillofacial) AND (anomalies)). A manual search of reference lists of relevant studies was also conducted.

The studies eligible for this review had to focus on patients, adults or children, suffering from rare genetic syndromes with dento-oro-craniofacial anomalies. Studies had to meet at least one of the following inclusion criteria:Rare genetic syndromes including dental anomalies.Rare genetic syndromes including anomalies of the maxillary bones.Rare genetic syndromes including anomalies of the soft oral tissues.Rare genetic syndromes including mixed oral anomalies.

Only papers written in English were considered.

Studies were excluded from the review if:They have focused on the genetic aspects only.

Book chapters and papers whose full-text was not available were not taken into consideration. From 789 papers obtained through PubMed and screened by title and abstract, 351 have been considered as eligible. After full text evaluation, 221 papers were definitely included.

## 3. Results

Sixty-two rare genetic syndromes involving oral manifestations were found in the 221 included papers (Figure 1).

Since no classification regarding oral abnormalities in rare genetic syndromes is present so far, the 62 rare genetic syndromes were classified by authors according to the involvement of bone, oral soft tissue, tongue, teeth, and mixed combinations of more than one structure (Table 2).

Five syndromes are described below via five different cases (in bold in Table 2), namely the Crouzon syndrome, a Congenital nemaline myopathy, the CHARGE syndrome, the Cornelia de Lange syndrome, and the Progeroid Syndrome.

### 3.1. Crouzon Syndrome

Twenty-eight papers describing the relationship between the Crouzon syndrome and oral anomalies were found in the PubMed search.

The Crouzon syndrome (CS) is an autosomal dominant genetic disease caused by mutations in one of the FGFR (fibroblast growth factor) genes, especially the FGFR2, which is responsible for the early closure of the cranial sutures. It has an incidence of 1 case in 25,000 births [18].

This syndrome has an extremely variable phenotypic manifestation in both the cranial and facial features [19,20]. On the one hand, some patients present a mild form of this syndrome that can guarantee a fully functional lifestyle. On the other hand, other patients suffer from a severe form of the disease that can cause a significant impact on their quality of life [20].

Clinical manifestations include tall and flattened forehead, maxillary hypoplasia and mandibular prognathism. For these reasons, this syndrome has been classified as a rare genetic syndrome affecting maxillary bones according to Table 2. The arrested growth of the upper maxilla causes bilateral severe proptosis and corneal exposure, thus resulting in shallow orbits. Hypertelorism, exorbitism, strabismus, and amblyopia are frequently found as well [21,22,23,24]. Alongside the ocular problems, these anatomical anomalies can lead to other functional problems like increased intracranial pressure and upper airway obstruction [25].

C. M. is a 10-year old male, patient of the Dental Clinic of San Paolo Hospital, Milan. During the extra-oral examination, the boy presented the typical signs of Crouzon Syndrome: a high and large forehead with a convexity in the region of the anterior fontanelle, flattening of the occipital region, bicoronal craniosynostosis and hypertelorism. He did not present maxillary hypoplasia or exorbitism. No mental retardation was observed.

The oral manifestations included the maxillary dental arch in V shape (Figure 2d), dental crowding, class II maloocclusion with an anterior severe open bite and lower labial interposition at rest (Figure 2a–c).

The intraoral examination showed an intermediate mixed dentition phase, decayed and filled teeth and gingivitis due to the presence of bacterial plaque, dental calculus, and oral breath (Figure 2).

Since the patient started an orthodontic treatment with upper and lower arch expanders in another dental clinic, the pre-treatment radiographs and pictures are not available.

The multidisciplinary treatment proposed consisted in: oral hygiene instructions, professional oral hygiene, 5% NaF varnish applications (every four months) and reassessment of orthodontic therapy already in place in order to continue increasing the transverse maxilla and mandibular width.

### 3.2. Congenital Nemaline Myopathies

Only one paper describing the relationship between congenital nemaline myopathies and oral anomalies was found in the PubMed search.

Congenital nemaline myopathies (CNM) are a heterogenous group of congenital myopathies caused by inherited mutations in more than twelve genes. CNM incidence is estimated to be one in 50,000 live births. The genes which encode skeletal α-actin (ACTA1) and nebulin (NEB) are the most frequently mutated [26].

The main manifestations of the disease include muscle weakness and hypotonia, with the presence of nemaline bodies in muscle fibers. CNM can manifest itself with multiple clinical phenotypes: in the most severe cases it can lead to neonatal death, while in the milder ones the disease causes only a slight impairment of motor function. When the respiratory muscles are involved as well, patients are ventilator-dependent [27]. No therapeutic treatment is available yet [26]. The clinical manifestations of CNM include severe muscle weakness, hypoventilation, swallowing dysfunction, and no speech ability. Oral or dental anomalies in patients with this disorder are not yet described in literature, except for a reduced development of the maxilla and mandible, due to muscular hypotonia [28]. For these reasons, this syndrome has been classified as a rare genetic syndrome affecting maxillary bones according to Table 2.

L.R. is a three-year-old boy affected by Congenital Nemaline Myopathy, who required a visit at the Dental Clinic of S. Paolo Hospital, Milan for severe mobility of all primary incisors.

The oral examination showed a severe reduced development of the maxilla, with a severe high-arched palate and reduced mouth opening (Figure 3).

Mobility of all the primary incisors was found, together with a delayed eruption of primary molars and canines. Due to the patient’s condition an orthopantomography was impossible to be carried out. Therefore, a periapical radiography was performed despite the difficulty in positioning the intraoral plate because of the reduced arches width (Figure 3b,c). A severe root resorption of unclear etiology and no evidence of permanent successors was noted (Figure 3). After consulting the patient’s cardiologist, the affected incisors were extracted under local anesthesia to avoid risk of swallowing or inhalation (Figure 3e).

### 3.3. CHARGE Syndrome

Twenty-one papers describing the relationship between the CHARGE syndrome and oral anomalies were found in the PubMed search.

The CHARGE syndrome is an autosomal dominant genetic disorder with multiple dysmorphic and congenital anomalies. It is a highly variable syndrome, usually diagnosed during the prenatal or neonatal period [29,30]. The actual incidence remains unknown, but it is believed to occur approximately in one case in 10,000–15,000 live births [31].

The term ‘CHARGE’ is an acronym generated by the initials of the most frequent clinical features: Coloboma, Heart defects, choanal Atresia, Retardation (of growth and/or development), Genitourinary malformation and Ear abnormalities [32]. Thus, the diagnosis is predominantly clinical, using the criteria proposed by Blake et al. [33], modified by Verloes [34], and then confirmed by genetic testing [31].

Clinical findings include orofacial cleft, distinctive facial appearance, tracheoesophageal fistula, limb abnormalities, cranial nerve dysfunction, semicircular canal hypoplasia, delayed attainment of motor milestones, genital hypoplasia, and rarely, immune deficiencies [29,30]. This syndrome has been classified as a rare genetic syndrome affecting maxillary bones and teeth according to Table 2.

A.T. is an eight-year old girl who attended the Dental Clinic of San Paolo Hospital, Milano for dental treatment. She was diagnosed both clinically (all the main signs of the CHARGE syndrome were present) and genetically, since the CHD7 gene mutation was confirmed. The girl presented bilateral chorio-retinal coloboma, cerebral-cerebellar malformation, middle and inner ear malformation, reduced weight and stature growth, globally delayed psychomotor development with absence of language. Cleft lip and cleft palate were also reported, surgically corrected in 2013. Intraoral examination showed a good oral hygiene and no dental caries lesions (Figure 4).

The intraoral examination showed a mixed dentition with lacking space in both arches for the eruption of the permanent incisors and a class III malocclusion with anterior crossbite (Figure 4c), due to maxillary hypoplasia.

An attempt to perform an orthopantomography was made, but the result was poor, and it was not possible to evaluate the presence of all permanent teeth. However, a prediction of dental crowding was detected.

The proposed treatment plan included follow-up visits every four months, semi-annually 5% Sodium-Fluoride varnish applications to prevent caries, and the application of sealants on permanent molars. Due to the patient’s low collaboration and poor oral hygiene, no orthodontic treatment has yet been planned.

### 3.4. The Cornelia De Lange Syndrome

Thirty-six papers describing the relationship between the Cornelia de Lange syndrome and oral anomalies were found in the PubMed search.

The Cornelia De Lange syndrome (CdLS) is a multisystemic disease characterized by growth retardation, intellectual and psychomotor impairment, abnormalities of the hands, hirsutism and facial dysmorphism [35]. Children with this disease present marked and arched eyebrows fused in the midline, thin lips and micrognathia [36]. The prevalence is about one case in 50,000 people.

Most cases are sporadic, although an autosomal dominant pattern of transmission has been identified and five genes seem mainly to be involved [37]. The NIPBL gene is altered in more than 50% of CdLS patients [38].

This syndrome is related to a series of alterations in the oral cavity: skeletal anomalies affecting both the maxilla and the mandible and dental problems. For these reasons, the Cornelia De Lange syndrome has been classified as a rare genetic syndrome affecting maxillary bones and teeth according to Table 2.

Regarding the skeletal anomalies, most CdLS-affected children present micrognathia, high-arched palate, and severe anterior open-bite with posterior occlusal contact. In addition, they present a class II dental and skeletal malocclusion, with maxillary prognathism and pro-inclination of the upper incisors. Due to the skeletal alterations, dental crowding generally occurs, in association with dental malposition [39,40].

About the dental aspects, many patients present dental agenesis, delayed eruption, and shape anomalies of roots and crowns. The presence of enamel defects is often detected, with areas of hypomineralized enamel on the vestibular surfaces. Moreover, some studies reported an increased prevalence of macrodontia and taurodontism. Therefore, in these patients, a high incidence of caries and periodontal problems are reported [41,42]. Furthermore, chin muscles contraction, hypotonicity of masticatory muscles, a short upper lip, ankyloglossia with lingual protrusion are also frequently found in CdLS-affected patients [40].

S.V. is a 12-year-old girl with CdLS, who attended the Dental Clinic of S. Paolo Hospital, Milan for dental treatment. The clinical examination showed a typical facial dysmorphism, bilateral conductive hearing loss, minor hand and foot anomalies and eyelid ptosis. She also presented severe growth, mental retardation, and speech disorders. Collaboration during dental visits and photos was very poor. The extraoral exam showed micrognathia and narrow upper maxilla (Figure 5).

The inspection of the oral cavity revealed dental crowding, dental malformations like mulberry molars, and tooth malposition with suspected premolars agenesis (Figure 6).

The patient also presented cleft palate surgically corrected with palatoplasty at 2 years of age. Dental caries, enamel defects, and gingivitis, due to poor collaboration and low level of oral hygiene, were also detected. An orthopantomography was performed during the first visit (Figure 6d), but the poor quality of the result due to the patient’s low degree of cooperation, reduced the exam reliability and made it impossible to confirm the premolars agenesis (Figure 6).

The treatment plan included oral hygiene sessions and topical applications of 5% fluoride varnish for caries control and prevention, restorative therapies, and extractions of decayed primary teeth to guide the eruption of permanent dentition, reducing dental crowding. Due to the poor collaboration and oral hygiene of the patient, no orthodontic treatment has yet been planned.

### 3.5. Progeroid Syndromes

Eleven papers describing the relationship between the Progeroid syndrome and oral anomalies were found in the PubMed search.

Progeroid syndromes are a group of rare genetic diseases, which share significant anomalies in the orofacial region. Progeria is characterized by a phenotype that mimics physiological aging.

The estimated prevalence is one case in 4 million people [43].

Progeroid syndromes are a group of disorders characterized by typical manifestations, which often include alopecia, atherosclerosis, lipodystrophy, diminished joint range of motion and scoliosis. Generally, these patients present a normal cognitive and motor development [44].

The oral apparatus is highly affected by progeroid syndromes. Patients have similar traits, which include low facial height, reduced thickness of the alveolar bone in the upper and lower jaws, dental anomalies of number, size and shape, enamel defects, and delayed eruption [45]. For these reasons Progeroid syndromes have been classified as rare genetic syndromes affecting maxillary bones and teeth according to Table 2.

M.C.T. is an eight-year-old girl affected by a progeroid syndrome of unclear genetic origin. The patient was submitted to a comparative genomic hybridization array, but no alterations were found. Nevertheless, she shared several characteristics with the Petty Syndrome. She was also diagnosed with hemangioma of the cephalic segment and of the back, scoliosis, skin xerosis, onychodystrophy and hypertrichosis. At the extra-oral examination, she also presented skin laxity and wrinkles of the neck and prominent eyebrows (Figure 7a–c).

The examination of the oral cavity showed a reduced thickness and height of the alveolar bones, both in the maxilla and in the mandible, a class III malocclusion in rest position and atypical swallowing with lingual interposition. Oligontia of primary and permanent dentition was also observed (Figure 7d–f). Because of these problems, the girl was used to eating only soft foods. No caries lesions were detected. Orthopantomography (Figure 8d) confirmed the oligodontia, delayed eruption and teeth shape anomalies, with short and thin roots (rizomicria). The condyles appeared dysmorphic and flattened.

The treatment proposed solution was oral hygiene sessions every 4 months and topical applications of fluoride varnish for caries prevention. The oral rehabilitation proposed was a total removable prosthesis for the upper arch and a partial removable prosthesis for the lower arch, using a plastic aligner on her natural teeth, with both mucosal and dental support (Figure 8a–c). This ad hoc prosthetic strategy was adopted since a normal anchorage was not possible due to the reduced crest dimension combined with the reduced stability of the permanent incisors (very short roots). The teeth 7.3 and 8.3, showing a third degree of mobility according to the Miller mobility index [46] (more than 1 mm of horizontal movement and depressible within the socket), were extracted, while 5.3 was left in situ since it did not cause problems in terms of rehabilitation. The child was instructed in the correct use and cleaning of the prosthesis. Follow-up visits showed an eruptive sprint of superior incisors (1.1 e 2.1) as they were stimulated by the prosthesis and multiple fracture of the partial removable prosthesis. During a follow up visit at the age of 12 years (Figure 9), the prosthesis of the upper arch was adjusted to endure teeth eruption, the lower arch prosthesis was removed, and a replacement was planned in accordance with the patient’s growth.

## 4. Discussion

As revealed in this narrative review, many rare syndromes involve dento-oro-maxillofacial anomalies, which are found not only in patients with syndromes affecting the ectodermal layer, but also in syndromes involving gene mutations that cause anomalies in districts far from the orofacial area (CNM, Progeroid syndrome, William syndrome, etc.).

So far, no classification of oral abnormalities in rare genetic syndromes is present in literature. The proposed classification aims to help the clinician to easily understand which dento-oro-maxillofacial findings might be expected in the presence of each rare genetic syndrome. This immediate framework may help in both diagnosing dento-oro-maxillofacial anomalies related to the underlying pathology and facilitating the drafting of treatment plans involving a multidisciplinary team.

Patients with genetic syndromes have generally well-defined clinical pictures. Although no papers describing the high prevalence of oral anomalies and diseases in patients affected by rare genetic syndromes are available, the assessed studies often report multiple oral problems [29,30,39,40,45,47].

This narrative review revealed only a few studies on each rare genetic syndrome, many of which are case reports, regarding the oral findings and the dental treatment carried out in affected patients. These data may be due both to the rarity of the syndromes taken into consideration, and to the poor interest reserved for the oral district compared to the general clinical conditions [48]. These factors, combined with the existing great heterogenicity among syndromes, weakened the possibility of providing general recommendations or guidelines on dental treatments of these patients.

Dental health provides a massive impact on quality of life and social acceptance. Several studies have shown a worse quality of life in patients with compromised oral health especially in children with special needs [49,50]. In addition, people with disabilities report difficulties accessing dental care and finding specialized dentists and barrier-free dental offices [51]. Oral prevention, namely treatment and maintenance programs associated with careful oral hygiene procedures carried out at home, is essential to prevent oral diseases, especially in patients with rare genetic syndromes. It is recommendable that such procedures are implemented as soon as possible [52]. A common treatment strategy adopted for all patients presented above was the application of fluoride varnish (5% NaF). Although caries is a multifactorial disease not related to genetic alterations, several factors may increase the risk of developing new caries lesions in subjects affected by rare genetic syndromes. Poor cooperation at the dental chair, difficulties in carrying out at home oral hygiene procedures, reduced chewing ability and drug intake are just some of the factors that can increase the risk of caries. For these reasons, the use of 5% NaF varnish, applied every three months, can reduce enamel demineralization increasing the strength of the dental tissue [53]. The treatment plan of these patients should aim to the restoration of fully functional teeth, jaws, and associated soft tissues. The dental recall intervals should be scheduled in rapid succession, according to one’s personal needs and risk of oral diseases [53]. The treatment plan should be chosen according to the level of cooperation of the patient/parent/caregiver [54], a fundamental factor to the success of the dental treatment.

Patients with the rare syndromes described in this narrative review (e.g., the Cornelia De Lange syndrome, the Crouzon syndrome) often have bone malformations. For this reason, an orthodontic assessment should always be included in the treatment plan, even if an orthodontic treatment is often not feasible due to the poor collaboration obtainable from these patients.

Dento-oro-maxillofacial anomalies should be intercepted as soon as possible, in order to prevent severe oral diseases (caries and periodontal disease) and the development of malocclusions, that may heavily affect oral functions and aesthetics. Treating the above reported patients, the authors noted a discrepancy between parental expectations and the actual needs of the patient: parents often require aesthetical improvements not in line with the therapies affordable and maintainable in a non-cooperative child.

Given the high complexity of the different clinical pictures described above, each treatment plan needs to be carried out by the collaboration of a team of professionals. The team should include neonatologists, pediatricians, pediatric neurologists, nutritionists, speech–language pathologists and therapists, maxillofacial surgeons, as well as pediatric dentists and orthodontics.

A limit of this review is the possibility that the proposed classification does not include all the relevant syndromes and related oral problems diagnosed nowadays. A wider search strategy could reveal additional syndromes showing dento-oro-maxillofacial anomalies. A strength of the present paper is instead represented by the proposed classification itself, that, as far as the authors know, is the first that tries to clarify the manifestations of the oral district in the complex clinical pictures of different rare genetic syndromes.

## 5. Conclusions

Over the years, the number of recognized rare genetic syndromes has been growing, along with the number of papers describing the organs and systems involved in each syndrome. In patients with rare genetic syndromes, oral findings and consequent dental treatments remain poorly investigated. The oral cavity is one of the most accessible organs involved in a rare syndrome, but also one of the most neglected. Dento-oro-maxillofacial anomalies and pathologies can benefit from simple and minimally invasive preventive and rehabilitative treatments, whose effectiveness is strongly related to the therapeutic timing. For these reasons, the proposed classification makes it possible to help clinicians to rapidly frame the different rare genetic syndromes on the basis of the type of dento-oro-maxillofacial anomalies that are capable of deeply compromising the quality of life of the affected patients.

## Figures and Tables

**Figure 1 children-09-00012-f001:**
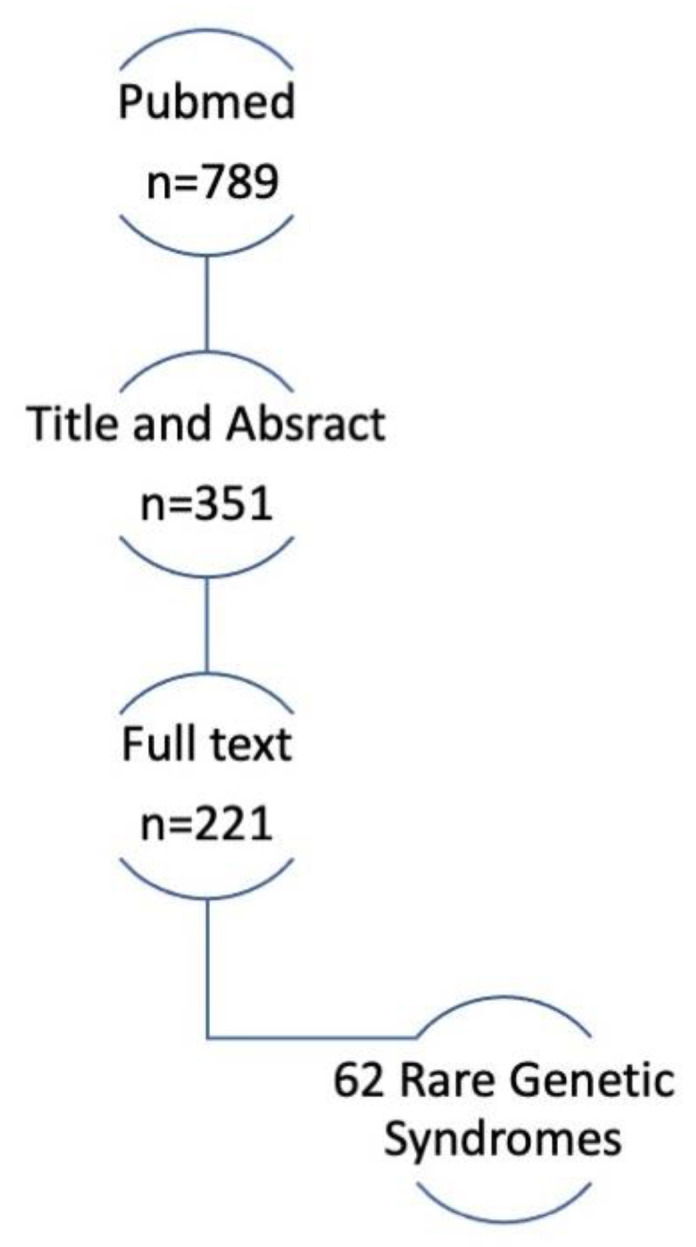
Flowchart.

**Figure 2 children-09-00012-f002:**
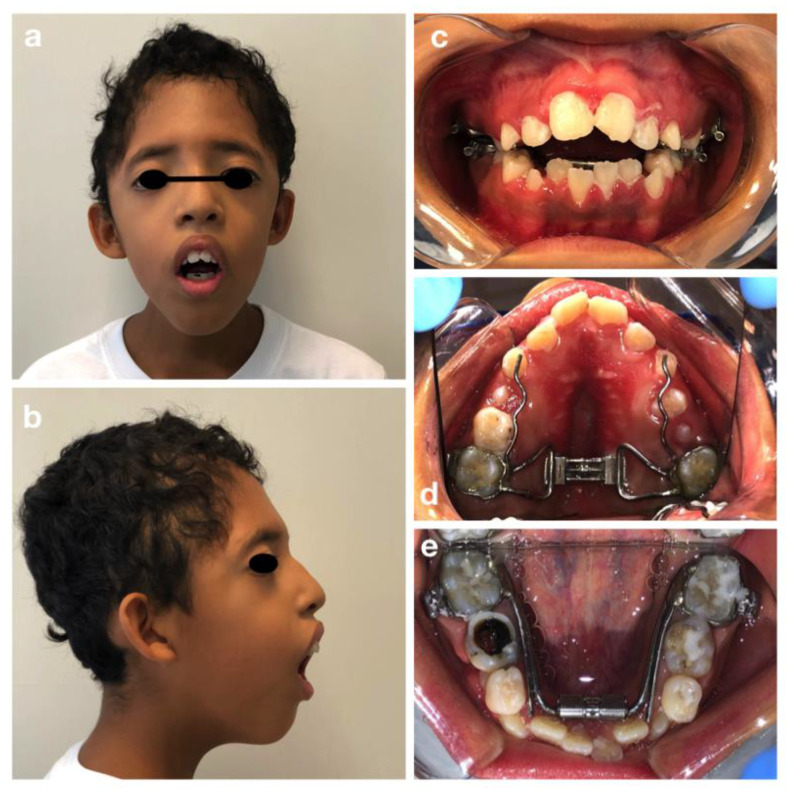
Crouzon syndrome: extra-oral (**a**,**b**) and intra-oral pictures (**c**–**e**). Upper and lower arch expanders are present (**d**,**e**). An arrested cavitated caries lesion in a second primary molar (8.5) is evident (**e**).

**Figure 3 children-09-00012-f003:**
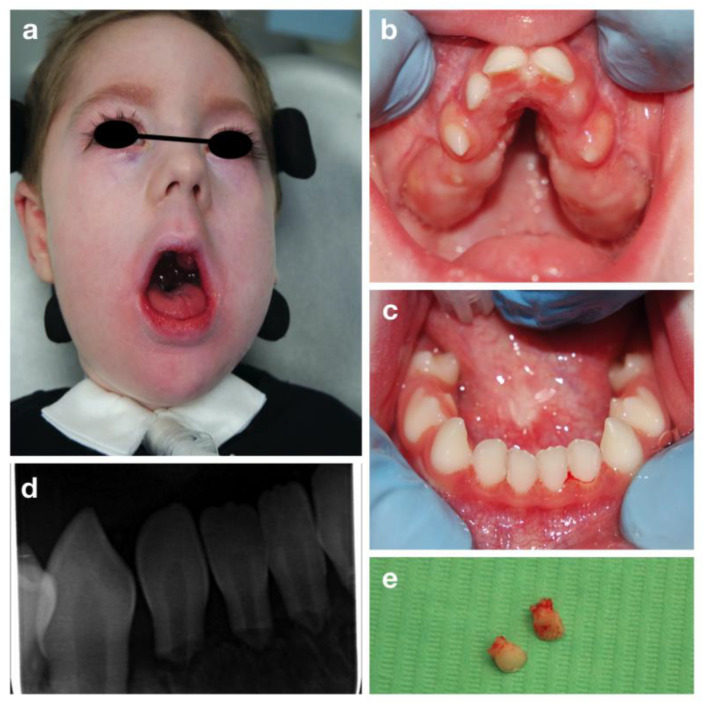
Congenital nemaline myopathies: extra-oral (**a**), intra-oral pictures (**b**,**c**), periapical radiograph of lower incisors (**d**) and the upper incisors after extraction (**e**).

**Figure 4 children-09-00012-f004:**
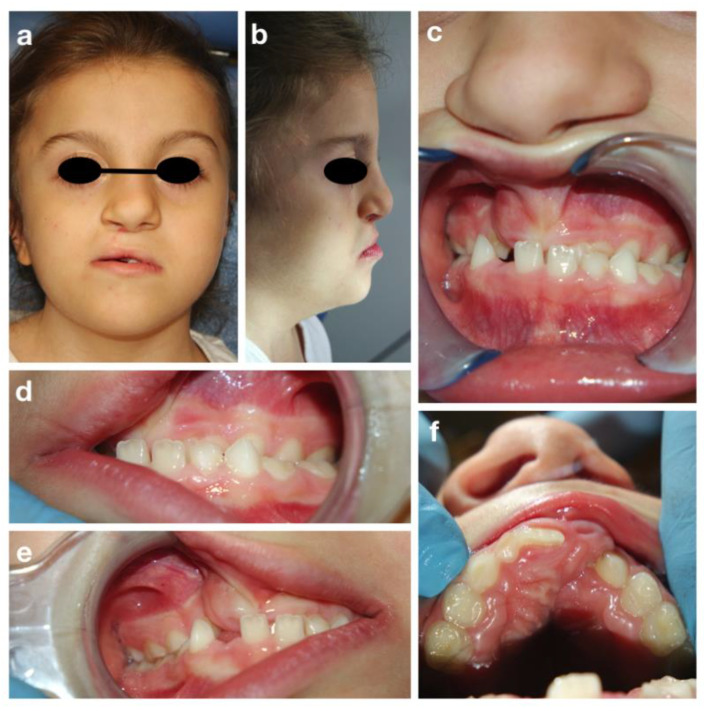
CHARGE syndrome: extra-oral (**a**,**b**) and intra-oral pictures (**c**–**f**).

**Figure 5 children-09-00012-f005:**
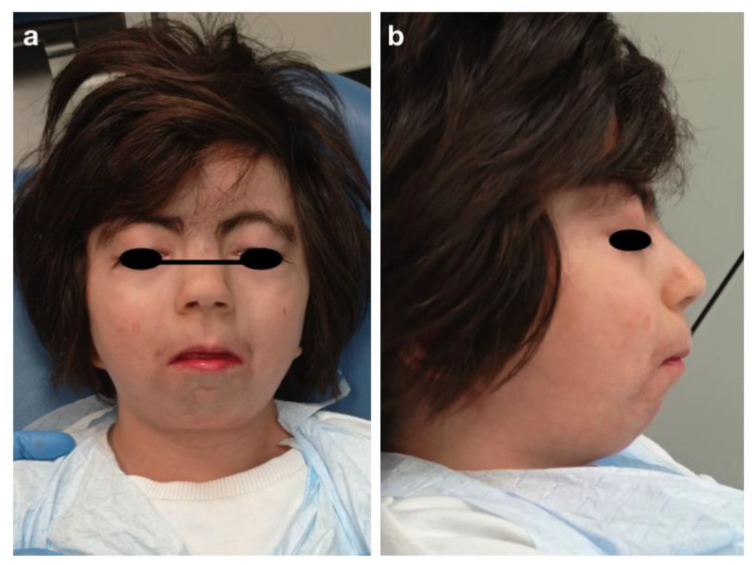
Cornelia de Lange syndrome: extra-oral pictures (**a**,**b**).

**Figure 6 children-09-00012-f006:**
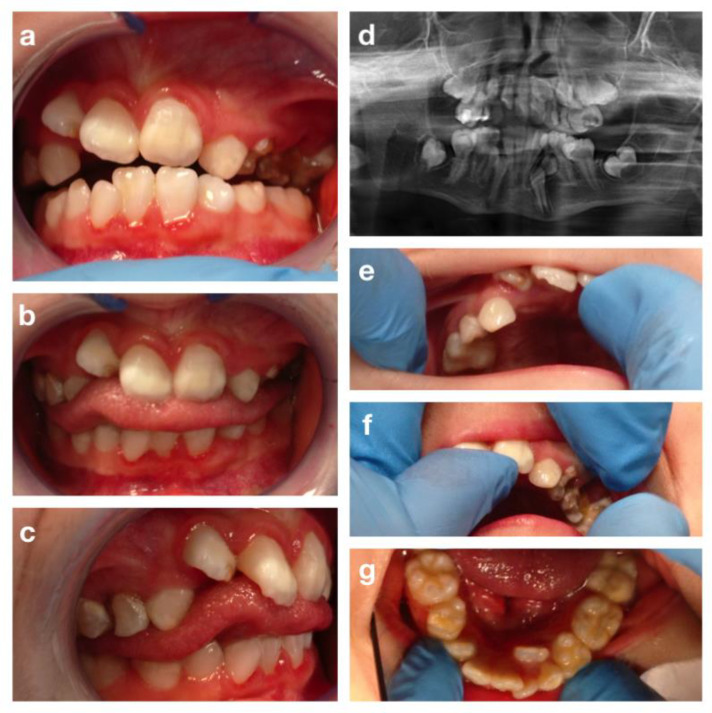
The Cornelia de Lange syndrome: intra-oral pictures (**a**–**c**,**e**–**g**) and orthopantomography (**d**).

**Figure 7 children-09-00012-f007:**
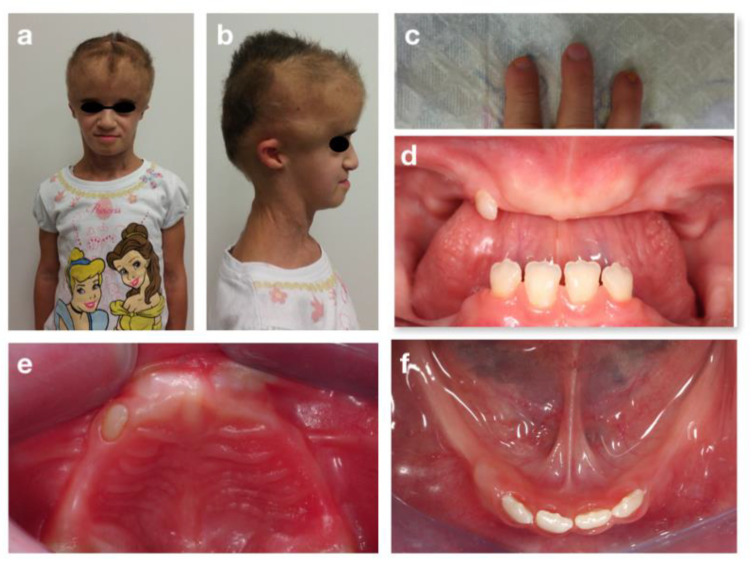
Progeroid syndrome: extra-oral (**a**,**b**), fingers (**c**) and intra-oral (**d**–**f**) pictures at the first visit.

**Figure 8 children-09-00012-f008:**
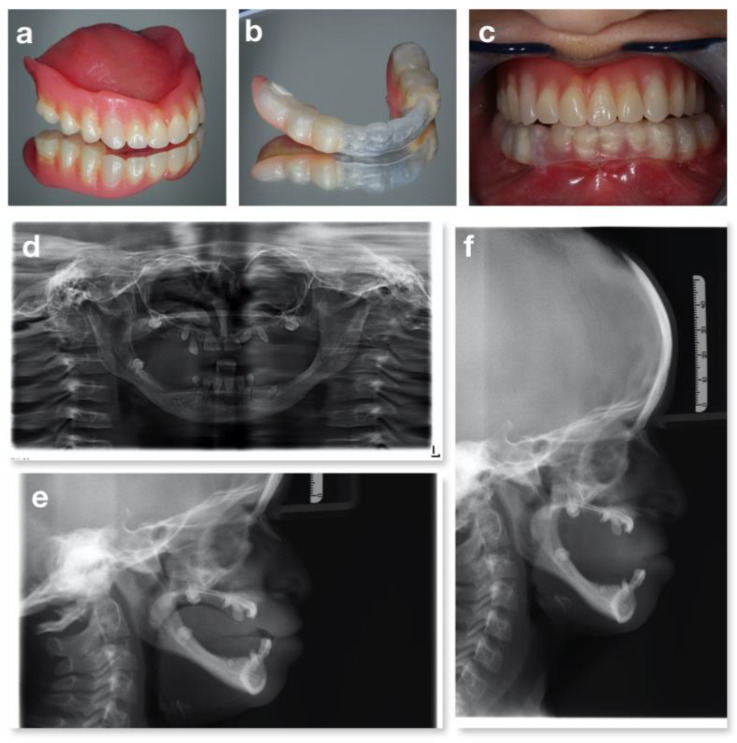
A Progeroid syndrome: prosthesis of the upper arch (**a**), prosthesis of the lower arch (**b**), intraoral occlusion with prosthesis (**c**), orthopantomography (**d**) and lateral teleradiography without (**e**) and with the total removable prosthesis in the upper arch and the partial removable prosthesis in the lower arch (**f**).

**Figure 9 children-09-00012-f009:**
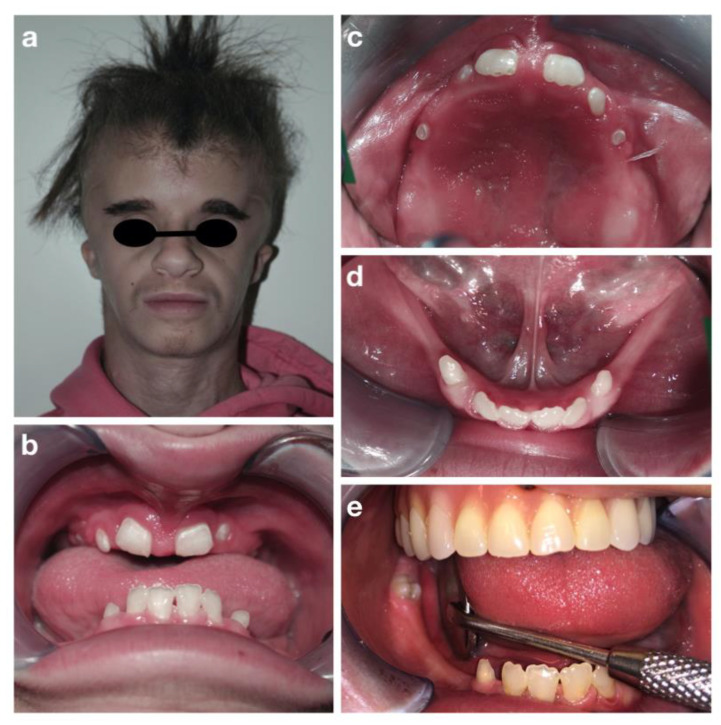
Progeroid syndrome: extra-oral (**a**), intra-oral (**b**–**e**) pictures at the follow-up visit at the age of 12 years.

**Table 1 children-09-00012-t001:** Simplified table of dental anomalies and oral warning signs.

Type of Anomaly	Clinical Signs	Anomalies Observed
**Number anomalies**		
**Reduced number**	Anodontia	Absence of all dental elements
	Oligodontia	Presence of less than half of the normal number of teeth
	Ipodontia	Presence of more than half of the normal number of teeth
**Increased number**	Supplementary teeth	The tooth repeats the shape and function of the adjacent tooth
	Supernumerary teeth	Elements are atypical, smaller and rudimentary. They are classified into:
		-Mesiodens: between the two upper central incisors
		-Paramolar: in the molar region
		-Distomolar: behind the third molar
**Location anomalies**	Ectopia	The element is located near the usual site, in a vestibular, lingual or palatal position
	Transposition	Two contiguous teeth reciprocally invert their position
	Heterotopia	Element located far from the usual position. If the position is outside the oral cavity it is defined as migration.
**Position anomalies**	Version	Inclination of a tooth towards the buccal or palatal side, forward (mesioversion) or backward (distortion)
	Inversion	Inverted tooth position (root versus alveolar ridge and crown versus basal bone portion). It is constantly accompanied by inclusion.
	Rotation	The element rotates along its longitudinal axis
	Intrusion	The tooth crown is on a lower plane than the occlusal plane
	Extrusion	The coronal margin is located higher than the occlusal plane
**Volume anomalies**	Gigantism	Macrodontia
	Dwarfism	Microdontia
	Taurodontism	Tooth with wide crown, short roots, extended pulp chamber
**Shape anomalies**	Crown anomalies	Accessory cusps, conical or tuberculate shape of the crown
	Roots anomalies	Anomalies in number, shape and size
	Endodontic anomalies	Anomalies in number, shape and size of the root canal morphology
**Developmental anomalies**	Enamel pearl	Small hard rounded excrescence, located near the enamel-cement junction.
	Fusion	Fusion of two teeth. Coalescence anomaly: due to close contact between germs; it can affect crowns and roots or only the crowns
	Concrescence	Two teeth are joined along the root surfaces by cementum. Coalescence anomaly
	Invaginated tooth	It is due to an infolding of enamel into dentine. There are two forms, coronal and radicular, with the coronal one being more common.
**Structural anomalies**	Enamel anomalies	Quantitative and/or qualitative enamel defect affecting some or all teeth. Amelogenesis imperfecta: abnormal formation of the enamel, unrelated to any systemic or generalized conditions. Autosomal dominant or autosomal recessive or x-linked pattern. It affects both deciduous and permanent.
	Dentin anomalies	Dentinogenesis imperfecta: brown teeth, crowns shrunk due to enamel flaking not supported by intact dentin. Autosomal dominant. It affects both deciduous and permanent.

**Table 2 children-09-00012-t002:** Classification of rare genetic syndrome based on the involvement of dento-oro-maxillofacial anomalies.

Dento-Oro-Maxillofacial Structures Involved	Syndrome
Maxillary bones	Basal cell nevus syndrome (Gorlin syndrome)
	Craniometaphyseal dysplasia
	**Crouzon syndrome**
	Fibrodysplasia Ossificans Progressiva
	Hyperparathyroidism-Jaw Tumor
	Mandibuloacral dysplasia with type A lipodystrophy
	Nager acrofacial dysostosis
	**Congenital nemaline myopathies**
	Osteopetrosis
Maxillary bones and teeth	Ablepharon syndrome
	Alagille syndrome
	Angelman syndrome
	Ankyloblepharon-ectodermal defects-cleft lip/palate (AEC) syndrome (type of Ectodermal Dysplasia)
	Cerebral, ocular, dental, auricular, skeletal anomalies (CODAS) syndrome
	**CHARGE syndrome**
	**Cornelia de Lange syndrome**
	Craniofrontonasal syndrome
	Dubowitz syndrome
	Ellis Van Creveld syndrome (Chondroectodermal dysplasia)
	Gardner syndrome
	Hallermann-Streiff syndrome
	Hutchinson-Gilford Progeria syndrome
	Hypophosphatemic rickets
	Kabuki syndrome (Niikawa–Kuroki syndrome)
	KBG syndrome
	Lenz microphthalmia
	Marfan syndrome
	Oculodento-osseous dysplasia
	Osteoglophonic dysplasia
	**Progeroid syndrome**
	Robinow syndrome
	Rubinstein Taydi syndrome
	Sanjad-Sakati syndrome
	Seckel syndrome
	Trichorhinophalangeal syndrome
Maxillary bones and tongue	Fraser syndrome
Maxillary bones, teeth and soft tissue	Floating-Harbor syndrome
	Larsen syndrome
	Lowe syndrome
	Rothmund-Thomson syndrome
	Silver Russel syndrome
Maxillary bones, teeth and tongue	Beckwith-Wiedemann syndrome
	Williams-Beuren syndrome (William syndrome)
Soft tissue	Gingival fibromatosis with hypertrichosis syndrome
	Kindler syndrome (type of Epidermolysis Bullosa)
	Pseudoxanthoma elasticum
	Recessive dystrophic epidermolysis bullosa of Hallopeau-Siemens
	Vascular Ehlers-Danlos syndrome
Soft tissue and maxillary bones	Zimmermann-Laband-1 syndrome
Soft tissue and teeth	Aagenaes syndrome/lymphedema cholestasis syndrome
	Prader-Willi Syndrome
	Rutherfurd syndrome
	Papillon-Lefèvre syndrome (type of Ectodermal Dysplasia)
Soft tissue and tongue	Hyalinosis cutis et mucosae
Teeth	Amelogenesis imperfecta
	Axenfeld-Rieger syndrome
	Dentin dysplasia
	Ectodermal dysplasia
	Focal dermal hypoplasia (Goltz syndrome)
	Jalili syndrome
	Oculo-facio-cardio-dental (OFCD) syndrome
	Witkop Tooth and Nail syndrome

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
