# Peer review of "Rare Genetic Syndromes and Oral Anomalies: A Review of the Literature and Case Series with a New Classification Proposal"

_children, 2021, doi:10.3390/children9010012_

Round 1

Reviewer 1 Report

The authors aimed to search through literature cases of rare geneticsyndromes affecting the cranio-facial structures (teeth, maxillary bones, oral soft tissues or mixed) and to propose a classification of oral abnormalities in rare genetic syndromes. In addition, five significant cases of rare genetic syndromes were presented.

The study  is well written, is easy to follow and covers an interesting topic. The methodology is well described with enough  data and results to support the work. The manuscript needs moderate grammar correction. Please also check typos thorough the text.

Conclusion Section: This paragraph required a general revision to eliminate redundant sentences and to add some "take-home message".

Author Response

Dear Reviewer thank you for your comments and suggestions. The paper has been reviewed according to your comment:

  • Typos and general grammar errors has been reviewed
  • Conclusion Section has been reviewed according to suggestions.

Reviewer 2 Report

Dear Authors,

I think this is a relevant narrative review, with detailed methodology and interesting results. Good goals, well planned, well developed. Paper is clearly written, easy to understand concepts and statements. I advise publication after minor revision.

Here are my comments for revision:

The abstract and the introduction section are well-done .  

The materials and methods section could be reorganized with subheadings. This section should be written in more detail. I suggest you to insert a flow chart with the diagram of the search, because it seems that 351 papers have been analyzed.

Based on what criteria were these 5 syndromes selected?  Motivate this choice in the text

Line 400: “Several 400 studies have shown a worse quality of life in patients with compromised oral health especially in children with Special Needs”. Please discuss more fully the issues that affect people with disabilities. Please consider:  doi: 10.3390/ijerph18041556

Author Response

Dear Reviewer thank you for your comments and suggestions. The paper has been reviewed according to your comment:

  • The materials and methods section has been written in more detail. A flow chart with the diagram of the search has been added as Figure 1.
  • The choice of the five syndromes selected has been motivated in the text (lines 147-154)
  • Line 464 "Several studies have shown a worse quality of life in patients with compromised oral health especially in children with Special Needs” has been discussed considering doi: 10.3390/ijerph18041556